# Prediction of Pollutant Concentration Based on Spatial–Temporal Attention, ResNet and ConvLSTM

**DOI:** 10.3390/s23218863

**Published:** 2023-10-31

**Authors:** Cai Chen, Agen Qiu, Haoyu Chen, Yajun Chen, Xu Liu, Dong Li

**Affiliations:** 1School of Geomatics and Urban Spatial Informatics, Beijing University of Civil Engineering and Architecture, Beijing 102616, China; chencai@casm.ac.cn (C.C.); 1108130422004@bucea.edu.cn (X.L.);; 2Chinese Academy of Surveying and Mapping, Beijing 100830, China; 3Jiangsu Provincial Surveying and Mapping Engineering Institute, Nanjing 210013, China; riong52178038@163.com; 4China Electronics Standardization Institute, Beijing 100007, China; chenyj@cesi.cn

**Keywords:** pollutant concentrations, residual network, ConvLSTM, spatial–temporal attention

## Abstract

Accurate and reliable prediction of air pollutant concentrations is important for rational avoidance of air pollution events and government policy responses. However, due to the mobility and dynamics of pollution sources, meteorological conditions, and transformation processes, pollutant concentration predictions are characterized by great uncertainty and instability, making it difficult for existing prediction models to effectively extract spatial and temporal correlations. In this paper, a powerful pollutant prediction model (STA-ResConvLSTM) is proposed to achieve accurate prediction of pollutant concentrations. The model consists of a deep learning network model based on a residual neural network (ResNet), a spatial–temporal attention mechanism, and a convolutional long short-term memory neural network (ConvLSTM). The spatial–temporal attention mechanism is embedded in each residual unit of the ResNet to form a new residual neural network with the spatial–temporal attention mechanism (STA-ResNet). Deep extraction of spatial–temporal distribution features of pollutant concentrations and meteorological data from several cities is carried out using STA-ResNet. Its output is used as an input to the ConvLSTM, which is further analyzed to extract preliminary spatial–temporal distribution features extracted from the STA-ResNet. The model realizes the spatial–temporal correlation of the extracted feature sequences to accurately predict pollutant concentrations in the future. In addition, experimental studies on urban agglomerations around Long Beijing show that the prediction model outperforms various popular baseline models in terms of accuracy and stability. For the single-step prediction task, the proposed pollutant concentration prediction model performs well, exhibiting a root-mean-square error (RMSE) of 9.82. Furthermore, even for the pollutant prediction task of 1 to 48 h, we performed a multi-step prediction and achieved a satisfactory performance, being able to achieve an average RMSE value of 13.49.

## 1. Introduction

The increasingly serious problem of air pollution has led to a great deal of societal anxiety in recent years [1]. Environmental management and the control of air pollution depend heavily on pollutant concentration predictions [2]. An essential element of air pollutants is PM_2.5_ (particulate matter with a diameter of less than 2.5 μm) [3]. Elevated concentrations of PM_2.5_ in the atmosphere pose a significant risk of respiratory infections, leading to diseases related to cardiopulmonary dysfunction, which are extremely detrimental to human health [4]. Predicting air pollution can provide the public and government agencies with effective early warning and support decision-making in response to serious pollution events [5]. Effective control of PM_2.5_ not only protects people’s health but also reduces social and economic losses. Therefore, accurate prediction of PM_2.5_ concentrations can provide timely early warning and enable governments to take timely action for the environment. Prediction of air pollutant concentrations, or simply air pollutant forecasting, plays an important role in air pollution prevention and environmental management [6], and thus it has recently received significant attention in the research community and is recognized as a key challenge in environmental management research.

PM_2.5_ concentration prediction can be viewed as a time-series processing problem that can be predicted based on past historical correlation data, e.g., meteorological factors such as temperature and humidity, as well as other pollution factors such as PM_10_ and O_3_ [7]. At this stage, the research methods are divided into two categories according to the characteristics of the research methods [8]: numerical model-based prediction methods and data-driven model-based prediction methods. The first is the prediction method of numerical modeling that simulates the process of emission, diffusion, transformation, and removal of air pollutants through meteorological principles and statistical methods so as to achieve the prediction of pollutant concentrations [9]; the second is the prediction method based on data-driven modeling, which is based on making predictions by learning and analyzing pollutant historical data [10].

Numerical modeling is mainly based on meteorological principles, knowledge of atmospheric dynamics and statistics, and the construction of equations for atmospheric pollutants and meteorological data to predict short-term pollutant concentrations [11,12]. Short-term predictions of pollutants generally refer to predicting pollutant concentrations for the next 1–6 h [12]. Then, according to the constructed atmospheric conditions, complex differential equations are solved by a computer to simulate the pollutants’ chemical, environmental, and transportation procedures throughout the atmosphere [13]. There are several commonly used numerical models: the Community Multiscale Air Quality Modeling System (CMAQ) [14] and the nested air quality prediction modeling system (NAQPMS) [15]. Despite the fact that numerical models take into account all changes in chemistry and atmospheric pollution transmission pathways, they suffer from the uncertainties of pollution sources, meteorological conditions, and transformation processes, as well as the high complexity of numerical models and the high arithmetic volume [16].

Compared with numerical modeling, the data-driven learning-based modeling approach is easy, effective, and universally feasible [6]. The model studies and evaluates historical information, focuses on mapping the relationship between historical data and air pollution concentration values in the predicted time period, and makes a more reasonable prediction of future pollutant concentration levels based on the current state [17]. Prediction methods based on data-driven models can be further subdivided into two categories: machine learning and deep learning models [11]. Machine learning models, an important type of artificial intelligence learning, combine the trends of the pollutants themselves and the intrinsic relationship between the pollutants and meteorology to produce predicted concentration values [18]. The pollutant concentration prediction models in common use today consist of Random Forest (RF) [19] models, Autoregressive Sliding Average (ARMA) models [20], and Support Vector Regression (SVR) [21]. These machine learning models can fully explore the nonlinear relationships between contaminant data with good robustness [22]. However, machine learning parameters generally rely on manual construction, which relies heavily on personal experience. In addition, they exhibit a lack of ability to reduce redundant data when dealing with larger and larger datasets, which then impacts their capacity for learning and generalization [23].

Deep learning models are more suitable to be applied within the field of predicting pollutant concentrations than traditional machine learning models [24]. Deep learning models can obtain better robustness through deeper hidden layers and excellent self-learning ability to explore higher-order nonlinear mapping relationships [25]. The following are some examples of deep learning models for time-based prediction: Recurrent Neural Networks (RNNs) [26], Gate Recurrent Units (GRUs) [27], and long short-term memory networks (LSTMs) [28]. In most cases, deep learning models outperform machine learning models in terms of effectiveness. Modeling the spatial–temporal interactions between numerous complicated nonsmooth air contaminants and meteorological data is necessary for air pollution prediction [29]. A single network structure may hinder the model’s ability to forecast accurately, all of which remain deficient when dealing with spatial–temporal big data.

Other deep learning techniques have been utilized by researchers to improve spatial–temporal modeling in order to address the limitations of single structure-based models. The following hybrid models have been employed for the prediction of pollutant concentrations: LSTM-FC [30], AC-LSTM [31], and EEMD-GRNN [32]. Due to shared weights and neighborhood recognition of convolutional processes, convolutional neural networks (CNNs) offer strong feature extraction capabilities [33]. As a result, using CNNs’ computer vision capabilities, CNN-LSTMs have effectively assessed the geographic dispersion features of atmospheric pollution concentrations [11]. We analyzed models based on CNNs to thoroughly examine the spatial and temporal correlations between pollutant data and meteorological parameters as a result of these studies [34].

Additionally, it is commonly acknowledged that attention processes can enhance the accuracy of predictions made by deep learning methods. In recent years, the ideas of attention processes and the processing of natural language and picture evaluation have become increasingly prominent [35]. Their major goal is to help models concentrate on their more crucial feature data. These are frequently employed in the prediction of time-series tasks related to, among other things, traffic, wind energy, and floods. For instance, to forecast flood events, Ding [36] designed a novel LSTM that mixes explainable temporal and spatial attention mechanisms (STA-LSTMs). Using spatial and temporal attention, weighting is given to the input data’s pattern of spatial–temporal properties. Additionally, STA-LSTMs fared better than the LSTM and CNN models. Nevertheless, the mechanism of spatial–temporal attention is yet to be investigated or used to forecast pollution. Undoubtedly, a method for bidirectional GRU combining attention was put forth by Zhang [37]. Studies have shown that the suggested model may outperform its competitors in capturing the most crucial aspects of historical data. Additionally, the prediction performance was significantly better than that of well-known RNN, LSTM and CNN-LSTM, although they did not consider spatial attentional mechanisms when trying to tap into spatial attributes. In order to improve air pollution prediction, it is worthwhile to investigate these potential spatial–temporal attention mechanisms in more detail [38].

In view of these, the secret to improving the effectiveness of pollution prediction is the efficient mining of spatial–temporal aspects of data. To address this issue and produce a more precise and reliable pollutant concentration forecast, this study provides a hybrid prediction model using spatial–temporal attention, ResNet, and ConvLSTM for pollutant concentration prediction. The contribution of this work is summarized below:(1)By using ResNet as the foundation layer of STA-ResConvLSTM, we avoid the problem of gradient vanishing or gradient explosion and provide for the removal of the deep network degradation issue and the extraction of spatially important information from meteorological and pollution data from numerous cities.(2)The spatial–temporal attention mechanisms are introduced into the residual block. Features in the temporal and spatial dimensions of pollutants are extracted using spatial–temporal attentional processes. As a result, the temporal and spatial dependencies can be effectively exploited, and the accuracy of pollutant concentration can be improved based on the weight distribution of attention.(3)ConvLSTM is used in the model as the final prediction layer. In order to fulfill the aim of mining the spatial–temporal correlation of the data, hidden advanced connection features must be extracted from the complex spatial–temporal sequence data generated via STA-ReNet. ConvLSTM avoids the gradient disappearing issue in addition to gaining from the effectiveness advantages associated with ConvLSTM with regard to time-series forecasting.

## 2. Data Description

### 2.1. Study Area

Beijing and its neighboring urban agglomerations were chosen to be this study’s subjects. The geographical distribution of the whole research region is depicted in Figure 1. With a large concentration of population and intensive social activities, the region is the political and economic center of China. Therefore, they were chosen as the main research objects. The industrialization process in the region is rapid, and industrial pollutants are emitted in large quantities. At the same time, the dense urban population and the diverse pathways of pollutant generation and diffusion have led to a serious situation of air pollution. In recent years, prevention and management of pollution in the atmosphere have been very effective, but compared to internationally advanced standards, there is still a lot of potential for improvement in the condition of the environment. The number of days where PM_2.5_ is the main pollutant accounts for nearly 38.9% of the total number of days that are contaminated per year [7]. Therefore, it is important to accurately predict PM_2.5_ concentrations in the region.

### 2.2. Data Characterization and Preparation

Multiple interactions exist between PM_2.5_, PM_10_, and other significant pollutants. In particular, PM_2.5_ and O_3_ have similar antecedents (such as NOx), and they likewise engage in a variety of atmospheric interactions [39]. As a result, the air quality index, also known as the AQI, and the six primary pollutants—namely, PM_2.5_, PM_10_, O_3_, NO_2_, CO, and SO_2_—were employed as inputs to the model. To simplify the structure of the inputs, it should be noted that the above factors employ the corresponding averages from numerous observation points in each city. Additionally, climatic factors like temperature, humidity, and wind speed can have an impact on how well pollutants in the atmosphere disperse [40]. As a result, we obtained hourly level information for the AQI and six key pollutants from the People’s Republic of China’s Ministry of Ecology and Environment (https://air.cnemc.cn:18007/) accessed on 21 April 2022. Daily weather information for the same time period was obtained through the Public Weather website (https://openweathermap.org/) accessed on 21 December 2022.

The gathered data underwent the following preprocessing: In this paper, the maximum number of missing data for Beijing pollutants in the data is 1782, the minimum number of missing data is 162, and the missing data rate is from 0.92% to 10.15%. Hence, the first step is to fill in the missing values in the dataset by simple linear interpolation [34]. As known from Table 1, the data values of PM_10_ in Beijing and Tianjin have the largest difference, and the data values of CO_24h have the smallest difference. The large difference in data values between different impact factors may increase the difficulty of modeling. In order to solve such problems, this paper uses the Min-Max function to standardize the data [38]. The following equation describes the Min-Max normalization formula:$$y^{\prime }=\frac{y-\min(y)}{\max(y)-\min(y)}$$
where min(y) denotes the minimum value of each factor and max(y) denotes the maximum value of each factor. Finally, the data were sequentially divided into a training set, a validation set, and a test set at a ratio of 70%:20%:10%. For a more illustrative presentation of the data, Table 1 presents the statistics for the two large cities of Beijing and Tianjin.

### 2.3. The Features of Data Distribution

#### 2.3.1. Exploration of the Temporal Dimension

Beijing’s yearly data for the year 2021 were chosen as this study’s target subject for the purpose of examining the features of how pollution concentrations and meteorological data are distributed. Figure 2 displays the yearly variation in value for each contaminant concentration. When additional pollutants and PM_2.5_ concentrations are observed, it is discovered that the pattern of those changes is often constant, which suggests that there might be an invisible connection among the pollutants. According to a statistical study, in 2021, PM_2.5_ concentration was to exceed the first intermediate limit of 35 $\mu g\cdot m^{-3}$, which has a slight negative influence on the well-being of some groups that are unusually sensitive. In 2021, the level of PM_2.5_ was to exceed 75 $\mu g\cdot m^{-3}$ 13.01% of the time, which was predicted to have an immediate impact on individuals’ everyday transportation and well-being [41]. Therefore, the prediction of PM_2.5_, for one, needs to take into account the occult connection between PM_2.5_ and other factors. Another aspect is that the study illustrated how timely prevention of the effects of high PM_2.5_ concentration on human wellness is possible.

The yearly value variations for each meteorological component are displayed in Figure 3. From Figure 3, first, it should be obvious that temperature changes are identical with variations in the barometric pressure, which are precisely the reverse of temperature changes. Second, it shows that there may be connections between the meteorological components because the values of the meteorological elements are extremely different yet their variations are very comparable. For example, as shown in Figure 3, high temperatures lead to low barometric pressure, and the reverse is also true. Third, it is obvious from the numerical curves of pollutant and meteorological data in one year that there are certain trends in meteorological and pollutant information. Therefore, for the purpose of addressing the longstanding pollutant concentration dependency, it is necessary to clarify the trend of pollutant and meteorological factors.

#### 2.3.2. Exploration of the Spatial Dimension

Figure 2 and Figure 3 show the meteorological and pollutant information within the temporal dimension, which is analyzed in detail in this paper. Beijing is a major city in the region. Its concentration of PM_2.5_ could additionally be described in the context of space. In a similar vein, this study chooses data on the concentrations of PM_2.5_ for every city in 2021. As shown in Table 2, this study calculates the concentrations of pollution and Pearson correlation coefficients among Beijing and nearby cities. Combining Table 2 and Figure 4, firstly, this paper observes that the correlation is higher for cities closer to Beijing, as indicated in bold. Second, when the distance between Beijing and other cities grows, the correlation values of pollutants in the atmosphere between Beijing and nearby cities steadily decline. The impact of distance hints at the spatial correlation of air contaminants.

The variations in concentrations of PM_2.5_ in Beijing and the other cities are then shown in Figure 4. First, it can be seen from Figure 3 and Figure 4 that the concentration of PM_2.5_ follows an overall trend that is lower when the temperature is elevated and elevated when the temperature is lower. Second, it has been discovered that the overall trends of PM_2.5_ vary across every city and are consistent in terms of the geographical and temporal dimensions. This illustrates the requirement of taking spatial correlation into account when making an effort to predict pollutant concentration based on the features of pollutant concentration correlation in Beijing and surrounding cities [42].

## 3. Methodology

### 3.1. Framework Overview

STA-ResConvLSTM was designed by us as the pollutant concentration prediction model, and its whole prediction procedure is a translation from the input of raw data to the output of results. The input of STA-ResConvLSTM is the past n hour historical data of pollutant and meteorological information ($x=\left\{x_{t},x_{t+1},\cdots ,x_{t+n-1},x_{t+n}\right\},x_{t+n}\in R^{s*i}$, where s indicates the quantity of observation stations and i indicates the total quantity of pollutant and meteorological factors). The output of the prediction model is the pollutant concentration in the next r hours ($\hat{y}=\left\{{\hat{y}}_{t},{\hat{y}}_{t+1},\cdots ,{\hat{y}}_{t+r-1},{\hat{y}}_{t+r}\right\}$, ${\hat{y}}_{t+r}\in R$, ${\hat{y}}_{t+r}$), which indicates the model’s predicted value for a given moment. Unlike other deep learning models for pollutant concentration prediction, a three-layer STA-ResConvLSTM prediction model is devised. The base layer of the model consists of an STA-ResNet, where each layer of the residual network performs convolutional operations on the input data using convolutional kernels of the same size and uses multiple convolutional layers to extrapolate spatial characteristics from the pollution and meteorological information provided. Adding a spatial–temporal attention module to the residuals block makes the model more attentive to the spatial–temporal characteristics of pollutant and meteorological information. At the conclusion of STA-ResNet, high-level spatial semantic characteristics $out=\left\{{out}_{t},{out}_{t+1},\cdots ,{out}_{t+n-1},{out}_{t+n}\right\}$ of pollutant and meteorological factor data are extracted by ResNet. The second layer is the ConvLSTM layer, which implements spatial–temporal feature extraction of pollutant concentrations. ConvLSTM uses a gating mechanism and convolutional operations. Like traditional LSTMs, the gating mechanism is employed for obtaining the data’s time-series characteristics, and the convolution operation is used to extract spatial features of the data. ConvLSTM successfully combines the data’s spatial–temporal features, making it possible to simultaneously extract these features. The last layer is the fully connected layer, which completes the final pollutant concentration prediction result $\hat{y}=\left\{{\hat{y}}_{t},{\hat{y}}_{t+1},\cdots ,{\hat{y}}_{t+r-1},{\hat{y}}_{t+r}\right\}$ after receiving the output from the ConvLSTM. The framework of our pollutant prediction model is shown in Figure 5.

### 3.2. Spatial–Temporal Attention

Attention mechanisms have their origins in human research in the visual domain. Humans consciously direct their restricted focus to the visually salient information while dismissing the unimportant data. Thus, the core task of the attention mechanism is to search for the internal relevance of the original data, thus ignoring irrelevant information and highlighting important information with a higher weight [43].

Given an original input $F\in R^{C\times H\times W}$ of a similar image, the dimensions *C*, *H*, and *W* mean the window size (indicating the magnitude of the historical observational input to the model), i.e., city information, air pollutant characteristics, and meteorological information, respectively. The schematic representation of the spatial–temporal attention module is shown in Figure 6. Since spatial–temporal attention has good weight distribution capabilities, this paper introduces a combination of spatial and channel attention to capture the spatial–temporal characteristics of meteorological and pollutant information [44].

#### 3.2.1. Spatial Attention

The pollutant and meteorological information are input into the spatial attention module of the STA-ResConvLSTM model in time-series order to assign different weights to the spatial features. Figure 7a shows a diagram of the spatial attention module.

First, on the input features, Maxpool(·) and Avgpool(·) operations are initially carried out. Then, a unique characteristic descriptor is created by connecting the outputs of two distinct characteristics. Finally, the convolution and sigmoid function procedures modify the new feature descriptors into new features. The formula for calculating SAM is as follows:(1)$$M_{s}(F)=f_{sigmoid}(Conv[MaxPool(F);AvgPool(F)]))$$
where Maxpool(·) indicates Maxpooling, Avgpool(·) represents averagepooling, the multi-layer perceptron is displayed by MLP(·), and Conv(·) displays the convolutional layer.

#### 3.2.2. Temporal Attention

The temporal attention mechanism, in contrast to the spatial attention mechanism, is more interested in the impacts of inputs from various historical pieces of information on the present and future. Figure 7b is the schematic representation of the temporal attention module. The temporal attention module is adaptive in acquiring the inner temporal correlations between the original inputs because the channel dimensions of the original input reflect the past time-lag information. By finding out each channel’s weights, the temporal attention module can enhance and suppress meaningful and useless historical information, respectively.

First, the spatial–temporal dimensions of the intermediate features are compressed by average pooling and maximum pooling to obtain spatial–temporal contextual features, respectively. After that, both of those characteristics are transformed by a shared multi-layer perceptron (MLP(·)) and merged using element-by-element summation. Finally, the merged features are activated by a sigmoid function to represent each channel’s importance weight. The computational procedure for the temporal attention module is as follows:(2)$$M_{c}(F)=f_{sigmoid}(MLP(MaxPool(F^{\prime })+MLP(AvgPool(F^{\prime })))$$

### 3.3. STA-ResNet

In this study, the spatial correlation characteristics of air contaminants and meteorological variables at numerous stations are extracted using ResNet’s intrinsic advantages. Meteorological and air pollution data are entered in time-series order in each residual block in an STA-ResNet. Then, the input data are processed by each ResNet convolutional layer to extract spatial features using the same convolutional kernel. Meanwhile, the extraction of the spatial and temporal weights of the initial inputs is carried out by the spatial–temporal attention modules accordingly. As a result, they enhance each other. The combination of both modules in each residual cell is shown in Figure 8. STA-ResNet extracted features are then output in temporal order $out=\left\{{out}_{t},{out}_{t+1},\cdots ,{out}_{t+n-1},{out}_{t+n}\right\}$. Each residual unit includes two convolutional layers and a spatial–temporal attention module. The residual unit is mainly divided into the residual part and the direct mapping part, and the formula is expressed as:(3)$$y_{i}=h(x_{i})+F(x_{i},w_{i})$$
(4)$$x_{i+1}=f(y_{i})$$
where ${h(x}_{i})$ is the direct mapping, F(·) represents the residual function, $w_{i}$ is the weight matrix, $f\left(\cdot \right)$ represents the Relu activation function, and $x_{i}$ and $y_{i}$ represent the input and output, respectively.

### 3.4. ConvLSTM Network

The STA-ResNet performs spatial characteristics extraction on pollutant and meteorological data to obtain time-series data $out=\left\{{out}_{t},{out}_{t+1},\cdots ,{out}_{t+n-1},{out}_{t+n}\right\}$ with high-dimensional spatial characteristics. This study takes advantage of ConvLSTM to perform spatial–temporal correlation feature extraction and pollutant concentration prediction on the output time-series data. In the process of spatial–temporal characteristics extraction, ConvLSTM handles spatial–temporal correlation characteristics extraction on the high-dimensional time-series data using gating mechanisms and convolution operations. In the process of pollutant concentration prediction, the fully connected layer receives each moment’s output states from ConvLSTM, which then generates pollutant prediction values based on the features of the extracted spatial–temporal correlation.

As shown in Figure 9a, we illustrate the comprehensive spatial–temporal feature extraction procedure using ConvLSTM in detail, where ($h_{t},c_{t}$) denotes the cell state. Each cell in the ConvLSTM has a special three-gate structure, where $i_{t}$ means the input gate, $f_{t}$ represents the forgetting gate, and $o_{t}$ means the output gate, which is similar to the LSTM. The ConvLSTM cell only differs in that convolutional functions are used for input-to-state and state-to-state transitions instead of fully connected operators. ConvLSTM outperforms LSTM greatly as a result of these enhancements. As shown in Figure 9b, the process of extracting spatial–temporal features using ConvLSTM can be described by the following equations:(5)$$i_{t}=\sigma (W_{xi}*x_{t}+W_{hi}*h_{t-1}+W_{ci}\circ c_{t-1}+b_{i})$$
(6)$$f_{t}=\sigma (W_{xf}*x_{t}+W_{hf}*h_{t-1}+W_{cf}\circ c_{t-1}+b_{f})$$
(7)$$g_{t}=tanh(W_{xg}*x_{t}+W_{hg}*h_{t-1}+b_{g})$$
(8)$$o_{t}=\sigma (W_{xo}*x_{t}+W_{ho}*h_{t-1}+W_{co}\circ c_{t}+b_{o})$$
(9)$$c_{t}=f_{t}\circ c_{t-1}+i_{t}\circ g_{t}$$
(10)$$h_{t}=o_{t}\circ tanh(c_{t})$$
where ∗ is the convoluted function, ∘ denotes the Hadamard product, tanh(·) is the TanHyperbolic function, and σ(·) is the sigmoid function. The output and state of the ConvLSTM unit at the previous instant are indicated by the variables $h_{t-1}$ and $c_{t-1}$, respectively. $x_{t}$ is the input of the current cell, and the potential memory cell for information transmission is $g_{t}$. The convolution kernels and bias terms are denoted by W and b, respectively.

### 3.5. Metrics

On the same dataset, this study’s suggested deep learning model is compared to other deep learning models. The root-mean-square error (RMSE), mean absolute error (MAE), coefficient of determination (R^2^), and index of agreement (IA) were used as metrics to prove the validity of the method. The computation formula is presented as is.
(11)$$\mathrm{RMSE}=\sqrt{\frac{1}{n}\sum_{i=1}^{n}{\left(y_{i}-{\hat{y}}_{i}\right)}^{2}}$$
(12)$$\mathrm{MAE}=\frac{1}{n}\sum_{i=1}^{n}\left|y_{i}-{\hat{y}}_{i}\right|$$
(13)$$R^{2}=\frac{\sum_{i=1}^{n}{\left({\hat{y}}_{i}-{\bar{y}}_{i}\right)}^{2}}{\sum_{i=1}^{n}{\left(y_{i}-{\bar{y}}_{i}\right)}^{2}}$$
(14)$$\mathrm{IA}=1-\frac{\sum_{i=1}^{n}{\left(y_{i}-{\hat{y}}_{i}\right)}^{2}}{\sum_{i=1}^{n}{\left(\left|y_{i}-\bar{y}\right|+\left|{\hat{y}}_{i}-\bar{y}\right|\right)}^{2}}$$
where n is the sample size of the input dataset of the model, ${\hat{y}}_{i}$ is the predicted value of the model, $y_{i}$ is the actual concentration of pollutants, and $\bar{y}$ is the average concentration of pollutants.

## 4. Results

### 4.1. Parameter Setting

The model’s structural layout and hyperparameter settings have a significant impact on the model’s ability to predict outcomes. We conducted several random search experiments to investigate the ideal hyperparameter values and structural architecture of the model in order to assure an equitable distribution of the model’s performance comparability. Table 3 displays the model test’s parameters.

### 4.2. Single-Step Prediction

This paper employs the great time-series data processing models CNN, LSTM, CNN-LSTM, and ConvLSTM as STA-ResConvLSTM model comparison models. Table 4 gives the quantitative results of the single-step prediction of PM_2.5_ concentration, comparing the differences between the CNN, LSTM, CNN-LSTM, ConvLSTM, and STA-ResConvLSTM models in terms of RMSE, MAE, R^2^, and IA. As can be seen from Table 4, the STA-ResConvLSTM model outperforms other deep learning models in the single-step prediction task of PM_2.5_ concentration. Compared with other comparative models, STA-ResConvLSTM improves R^2^ to 0.9307, IA to 98.29%, RMSE to 9.82, and MAE to 5.86, which significantly improves prediction accuracy. In addition, the prediction performance of single-structured deep learning models (CNN and LSTM) is significantly lower than that of hybrid deep learning models (CNN-LSTM and ConvLSTM). This means that the hybrid deep learning model outperforms the single-structured deep learning model in the single-step prediction of PM_2.5_ concentration. Compared to the CNN-LSTM model, STA-ResConvLSTM improved R^2^ by 0.041, IA by 0.87%, RMSE by 1.77, and MAE by 0.47. Compared with the CNN-LSTM model, STA-ResConvLSTM improves R^2^ by 0.041, IA by 0.87%, RMSE by 1.77, and MAE by 0.47. This is because STA-ResConvLSTM can learn spatial–temporal correlation characteristics of pollutants and meteorological information through spatial–temporal attention. Compared with CNN and LSTM, ResNet and ConvLSTM can obtain deeper spatial–temporal features of PM_2.5_ concentration, which can improve the final prediction results.

### 4.3. Multi-Step Prediction

Pollutant concentration prediction research has mostly concentrated on single-step prediction; however, this is insufficient to fulfill the demands of daily life. Predicting pollutant concentration over an extended period of time in future periods is the goal of multi-step prediction [7], and its forecast might serve as a helpful guide for travelers. In this section, the performance of the models for multi-step prediction of pollutant concentrations is analyzed. Table 5 shows the quantitative results of the multi-step prediction of PM_2.5_ concentrations. The RMSE of STA-ResConvLSTM decreases to 12.63 and the MAE to 8.52. The R^2^ of STA-ResConvLSTM increases to 0.8871 and the IA to 97.19%, which is a significant improvement in prediction accuracy when compared with the other comparative models. Moreover, the error of the ConvLSTM model is lower than that of the CNN-LSTM model, but the difference between the two is not significant, and both values are lower than that of the deep learning model with a single structure. As a result, the hybrid deep learning approach is superior to the traditional deep learning model for the multi-step prediction problem of PM_2.5_ concentration. In addition, compared with the ConvLSTM model, STA-ResConvLSTM improved R^2^ by 0.0657, IA by 1.31%, RMSE by 1.8, and MAE by 0.28. The results show that ConvLSTM, with its excellent spatial–temporal feature extraction of pollutants, has higher prediction performance than the other models. However, the STA-ResConvLSTM model performed better than the ConvLSTM model in both single-step and multi-step prediction.

Everyone is aware that model prediction becomes more difficult as the prediction time step increases [8]. This section investigates the effect of the prediction time step on the model constructed in this study and other deep learning models. To assess the model created in this research and other deep learning models for PM_2.5_ concentration’s multi-step prediction capabilities, the quantitative results of the prediction are presented in Figure 10 through the change curves of RMSE, MAE, R^2^, and IA. As shown in Figure 10, all prediction models’ accuracy declines as the prediction time step grows. This is because, as the prediction time step increases, the difficulty of prediction increases and the accuracy of the model decreases. Along with the process of increasing the prediction time step, the prediction performance of the STA-ResConvLSTM model decreases slowly and tends to a stable state, e.g., the RMSE stabilizes at about 12.5 and the R^2^ stabilizes at about 0.88. It is observed from Figure 10 that the forecasting accuracy values of CNN-LSTM and ConvLSTM are nearly identical, as are the forecasting accuracy values of LSTM and CNN. As shown, the four metrics used to evaluate the performance of CNN-LSTM and ConvLSTM consistently outperform CNN and LSTM. This shows that, as the prediction issue becomes more challenging, the hybrid structural model may characterize complicated data more accurately. CNN-LSTM’s prediction performance cannot outperform that of STA-ResConvLSTM in any time period. This indicates that spatial–temporal attention, ResNet, and ConvLSTM bring stability to the model’s prediction and can better handle complex spatial–temporal data in multi-step predictions. In addition, it is observed from the figure that, in comparison to other models with various prediction time steps, STA-ResConvLSTM has the lowest prediction error and the best forecast accuracy.

### 4.4. Trend Prediction

To ensure that the trend forecast made by the model developed in this study is stable, the pollutant and meteorological information data from the previous 24 h were constructed as three-dimensional inputs for the prediction model. These inputs were utilized to forecast the trend of PM_2.5_ concentration in the following 48 h. The STA-ResConvLSTM constructed in this paper was compared with other PM_2.5_ prediction models, and Table 6 and Table 7 show the changes in RMSE, MAE, R^2^, and IA of CNN, LSTM, CNN-LSTM, and ConvLSTM with the model in this paper. As shown in Table 6 and Table 7, the model constructed in this paper can continue to significantly outperform other prediction models as the prediction time step increases. Furthermore, the four performance evaluation indicators of the CNN and LSTM models fluctuate greatly in long-term predictions. The four evaluation indexes of this paper’s model fluctuate less with respect to the prediction time step (the smallest change in value), indicating that STA-ResConvLSTM is the most suitable choice for the multi-step prediction of PM_2.5_ concentration. The STA-ResConvLSTM model performs well in predicting pollutant concentrations at a time step of 48 h with less fluctuation in accuracy, which suggests that the model can continue to predict pollutant concentrations for longer time periods. But as the prediction step increases, the accuracy of the prediction decreases. CNN-LSTM has more spatial feature extraction ability than LSTM; the results in the table prove that CNN-LSTM is more suitable for time-series prediction tasks than LSTM. However, it is far from enough to obtain spatial characteristics of regional pollutant concentrations only with CNN. CNN cannot filter unimportant information in pollutant and meteorological data, and it is challenging to obtain an in-depth spatial–temporal correlation between pollutants and meteorological counts in the region. Therefore, in this paper, temporal attention, ResNet, and ConvLSTM are combined into a new CBAM-ResConvLSTM model to fully utilize the advantages of their components.

To further validate the prediction performance of the suggested model for pollutant trend changes, this paper analyzes the fitting ability of models for predicting PM_2.5_ concentrations with a time step of 48 h. Figure 11 shows a line graph and scatter plot of the change in predicted and true PM_2.5_ values over the next 744 h (1 December 2021, 0:00 to 31 December 2021, 23:00). As shown in Figure 11, it can be seen that CNN makes the worst predictions and is unable to interpret the PM_2.5_ concentration trend. Compared with LSTM, it is more capable of predicting PM_2.5_ concentration, but the prediction accuracy of the sudden change point (sudden change in value over a short period of time) in PM_2.5_ concentration is not enough. Although the ConvLSTM curve fluctuation is not very large, it is challenging to forecast the trend in PM_2.5_ concentration. The model constructed in this study outperforms other comparative models in predicting the sudden change point in PM_2.5_ concentration and the change trend in PM_2.5_ concentration. When the concentration of PM_2.5_ pollution sources is unstable (when PM_2.5_ concentration is greater than 60 $\mu g/m^{3}$), the traditional deep learning model cannot capture the real change trend and presents very confusing results. This illustrates the fact that the model’s ability to forecast future PM_2.5_ concentrations accurately is still limited. In addition, the STA-ResConvLSTM prediction results are basically consistent with the observation results, which means that the model constructed in this paper has a very good fitting result on the prediction of the mutation points and the change trends in PM_2.5_ concentration.

As a result of combining every model’s ability to match the data in Figure 11, the following conclusions may be drawn: (1) The experimental results confirm that for single-step, multi-step, and trend prediction of pollutant concentration, the STA-ResConvLSTM model predicts the trend of pollutant concentration with a very strong reference value. (2) From Figure 11(a1–e1), the STA-ResConvLSTM model’s prediction performance is superior to that of the comparison model, and it is appropriate for the purpose of predicting the abrupt change in pollutant concentration. (3) As can be observed in Figure 11(a2–e2), the STA-ResConvLSTM model is able to forecast high concentrations of PM_2.5_ more correctly than the comparison model. High agreement exists between the projected and actual values. (4) It is intuitively obvious when paired with the experimental findings in Figure 11 that the concentration of PM_2.5_ is often greater and the total number of mutation points is smaller. This mostly reflects the issue that there are not many samples at the mutation locations in the overall dataset, which causes an issue with unequal data distribution. The occurrence causes an issue of inadequate learning in the prediction models, which makes it challenging to learn the pattern of change in pollutant concentration in the event of a mutation. Because of this, certain models might be challenging to fit when there is an abrupt increase in pollutant concentration.

The STA-ResConvLSTM model can predict pollutant concentrations beyond the next 48 h in a multi-step prediction, but the prediction performance becomes unstable with the prediction time step.

## 5. Discussion

The results show that STA-ResConvLSTM has the best performance among all the tested models for single-step, multi-step, and trend prediction of PM_2.5_. The hybrid deep learning framework based on spatial–temporal attention mechanisms becomes a more useful tool for processing spatial–temporal data than its deep learning model.

From the temporal dimension, there is a clear cyclical variation in pollutants and meteorological data, which can also be said to be time-dependent. From the spatial dimension, the PM_2.5_ values of the ten cities are similar, and it can also be said that the pollutants have a spatial correlation.

From the results of the PM_2.5_ single-step and multi-step prediction experiments, it can be seen in Table 2 and Table 3 that CNN-LSTM, ConvLSTM, and STA-ResConvLSTM have better prediction results compared to the CNN and LSTM methods because all three methods can handle pollutant prediction problems. Next, comparing the prediction results of CNN, CNN-LSTM, and ConvLSTM in Table 2 and Table 3, it can be concluded that the prediction accuracy of ConvLSTM is higher than that of the other models, which proves that ConvLSTM has a better ability to extract spatial features of pollutants and meteorological data. Finally, comparing the prediction results of ConvLSTM and STA-ResConvLSTM in Table 2 and Table 3, it can be seen that the prediction accuracy of STA-ResConvLSTM is higher than that of ConvLSTM, which proves the superiority of the spatial–temporal attention mechanism and residual network for deep feature extraction of spatial–temporal data. The experimental results of the STA-ResConvLSTM model in Table 2 and Table 3 also confirm that it is very effective for the prediction of PM_2.5_. The optimal values of RMSE are only 9.82 and 12.63, respectively.

From the results of the PM_2.5_ trend prediction experiment, as shown in Figure 11 for CNN and LSTM, the curve fluctuates greatly, and it is difficult to predict the trend in PM_2.5_ concentration. CNN-LSTM’s and ConvLSTM’s curves fluctuate little and are stable, but it is difficult to predict the trend in PM_2.5_ concentration at the sudden change point. Combining Table 6 and Table 7 and Figure 11, compared to other deep learning models, STA-ResConvLSTM has the least fluctuation in prediction accuracy with the increase in prediction time step and can accurately predict the future trend in pollutant concentration. From the figure, we can see that the trend of the observed and predicted curves in the red box is consistent. Therefore, in the future pollutant prediction process, the STA-ResConvLSTM model can be considered to be combined with state-of-the-art prediction methods so as to improve the accuracy of pollutant prediction more effectively.

## 6. Conclusions and Future

This paper reports a model for pollutant concentration prediction. First, the design of spatial–temporal inputs to the model is guided by a correlational examination of inter-city pollution and meteorological information. Once more, using the concepts of spatial–temporal big data correlation combined with deep learning, an STA-ResConvLSTM prediction model based on spatial–temporal attention mechanisms ResNet and ConvLSTM is constructed. The framework is mostly employed to forecast the concentrations of pollutants in target cities. To explore the spatial–temporal dependency of historical knowledge, one uses spatial–temporal attention. The primary function of ResNet is to obtain the spatial characteristics of meteorological and pollution data from various cities. The high-dimensional information generated from ResNet is processed using ConvLSTM to obtain the spatial–temporal characteristics. The benefits of the suggested technique are outlined in the list below:(1)The temporal attention mechanism and spatial attention mechanism enable the model to capture more spatially and temporally dependent important information than other prediction models.(2)Compared with traditional CNN, ConvLSTM, and CNN-LSTM, ResNet can better extract spatial characteristics with the same deep network.(3)The prediction model presented in this study uses the ConvLSTM as the output layer due to the spatial–temporal correlation of atmospheric pollutants. Compared with LSTM, ConvLSTM extracts the hidden high-level correlation features in the 3D data to realize the goal of mining the spatial–temporal correlation of the data.

Although the superiority of the designed prediction model has been well established, there are still deficiencies that require further improvement. One approach is to divide pollutant and meteorological data from different cities into grids to extract spatial information more efficiently. In addition, we plan to obtain pollutant and meteorological information for more cities or longer time spans, as more information is anticipated to increase the accuracy of the model.

## Figures and Tables

**Figure 1 sensors-23-08863-f001:**
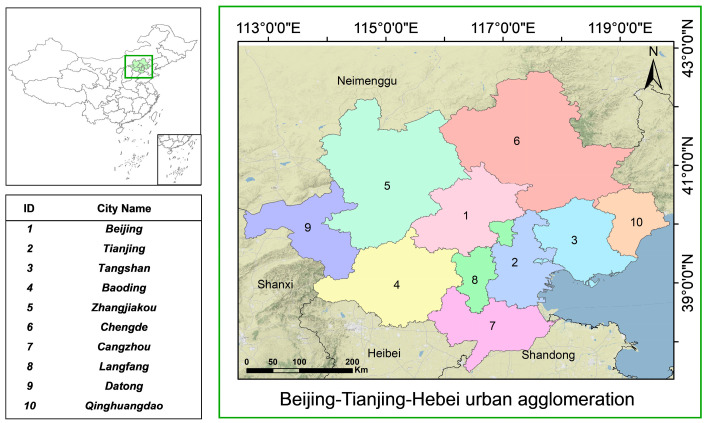
The geographic distribution of the study area.

**Figure 2 sensors-23-08863-f002:**
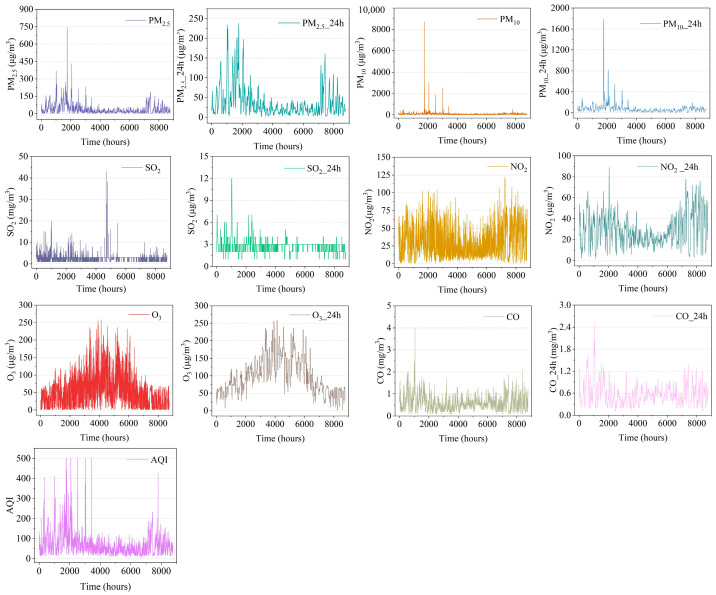
Time-series plots of air pollutant concentration data.

**Figure 3 sensors-23-08863-f003:**
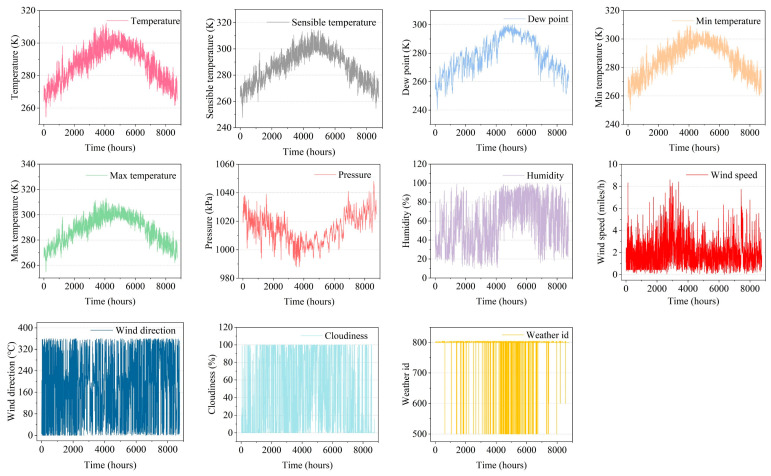
Time-series plots of meteorological data.

**Figure 4 sensors-23-08863-f004:**
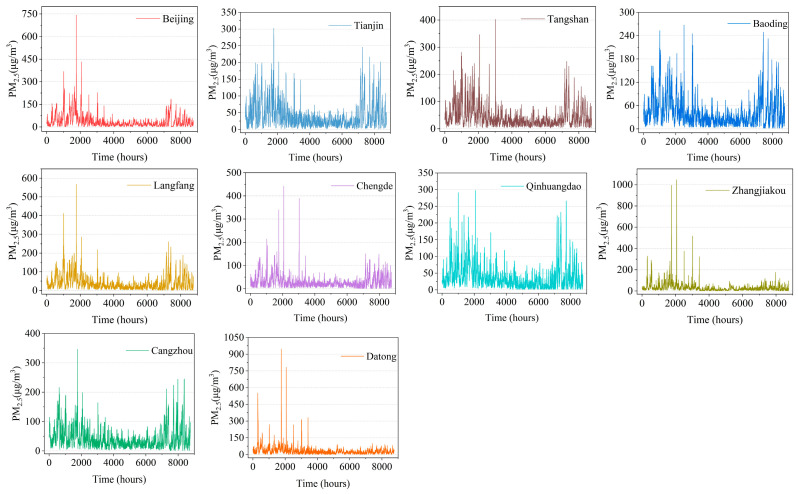
Spatial distribution characteristics of PM_2.5_ in Beijing and neighboring cities.

**Figure 5 sensors-23-08863-f005:**
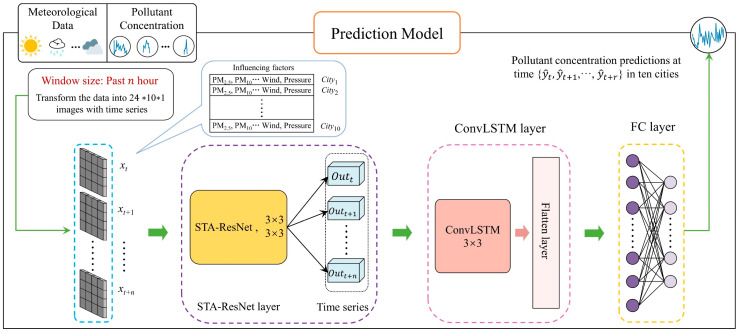
Framework of the STA-ResConvLSTM model for pollutant concentration prediction.

**Figure 6 sensors-23-08863-f006:**
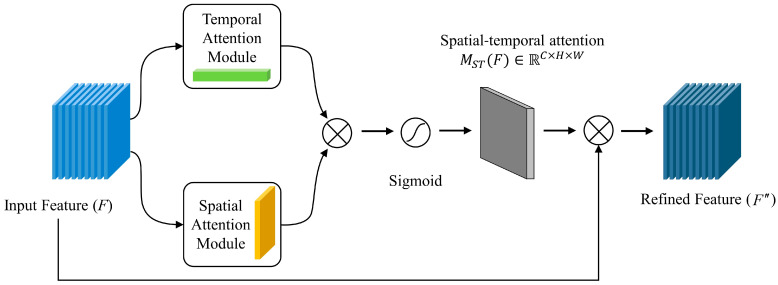
Framework of spatial–temporal attention module.

**Figure 7 sensors-23-08863-f007:**
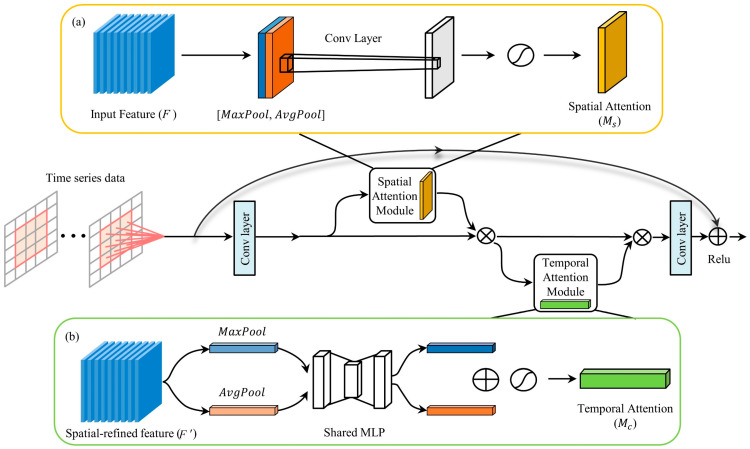
Framework of spatial attention module and temporal attention module. (**a**) Spatial attention module. (**b**) Temporal attention mechanism.

**Figure 8 sensors-23-08863-f008:**
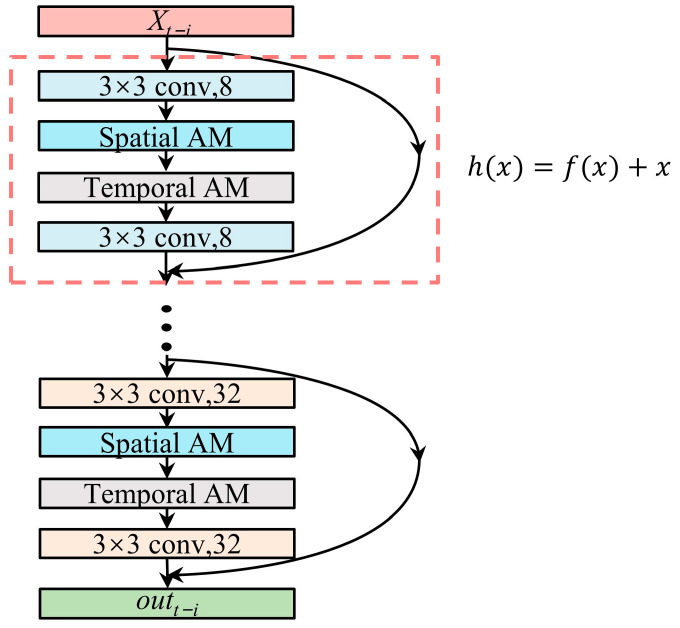
Framework of STA-ResNet.

**Figure 9 sensors-23-08863-f009:**
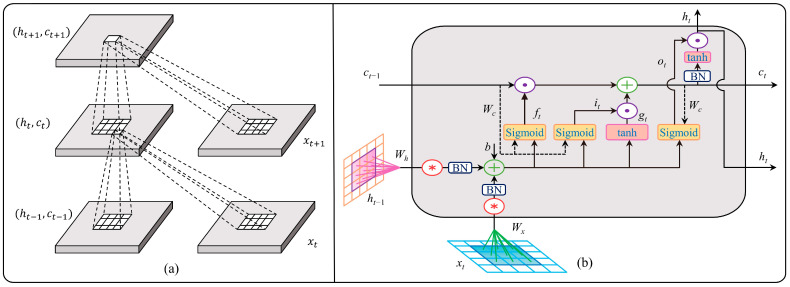
Framework of ConvLSTM. (**a**) Spatial–temporal feature extraction process of ConvLSTM. (**b**) One-cell structure of ConvLSTM.

**Figure 10 sensors-23-08863-f010:**
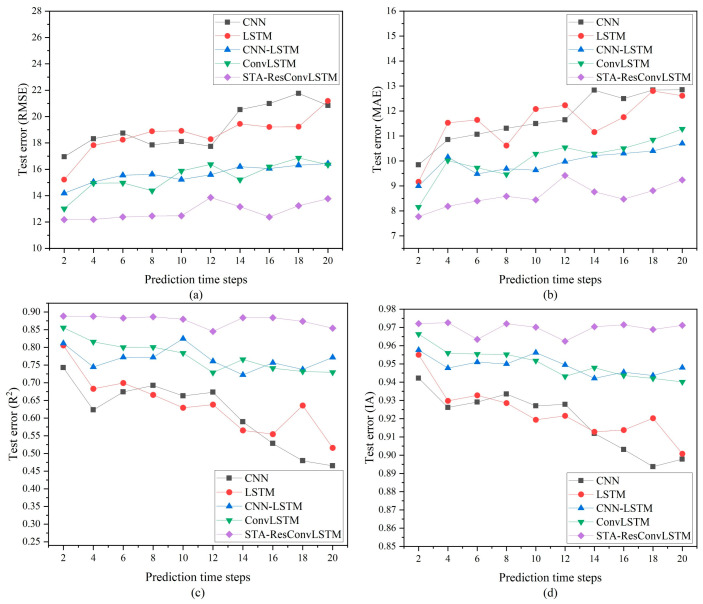
RMSE, MAE, R^2^, and IA of prediction model at different prediction time steps. (**a**) RMSE. (**b**) MAE. (**c**) R^2^. (**d**) IA.

**Figure 11 sensors-23-08863-f011:**
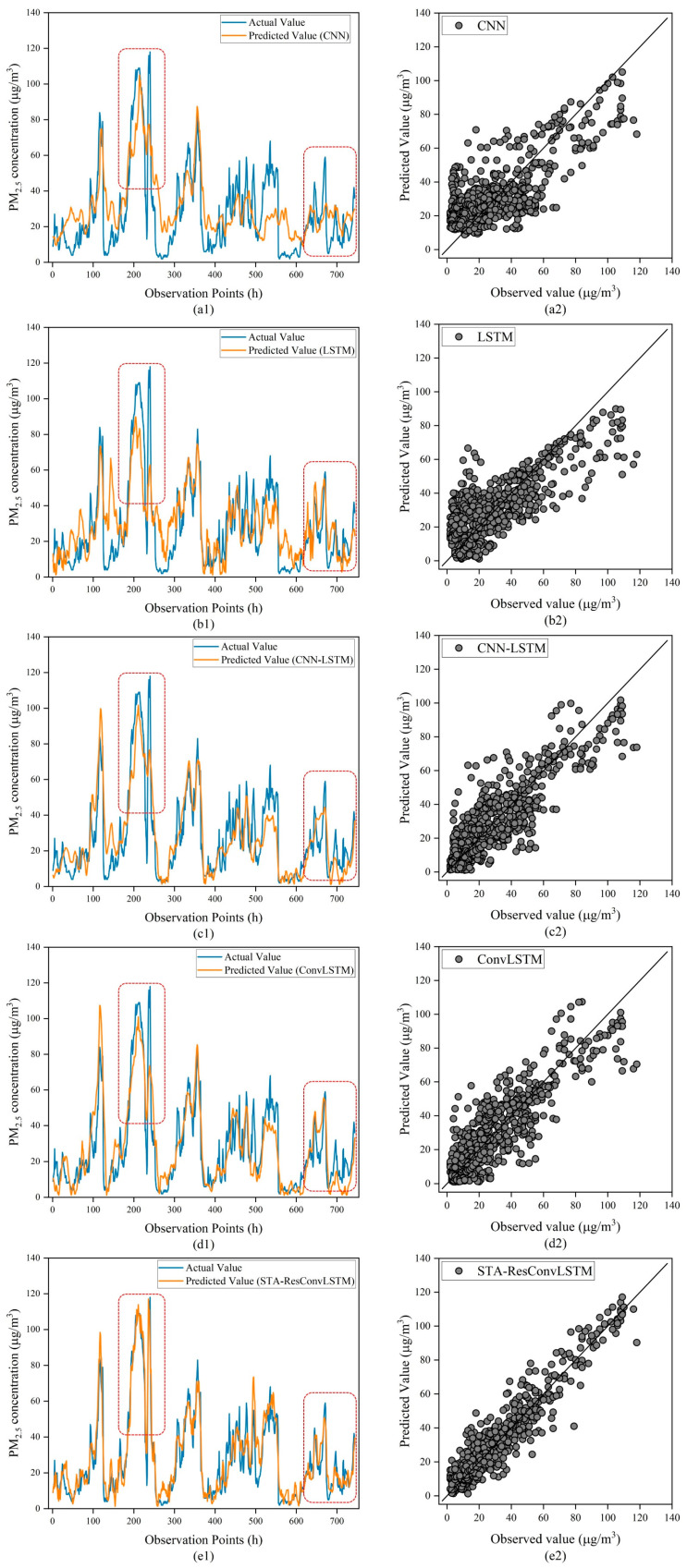
Comparison of PM_2.5_ concentration prediction models for Beijing. (**a1**) Line graph of CNN. (**a2**) Scatter plot of CNN. (**b1**) Line graph of LSTM. (**b2**) Scatter plot of LSTM. (**c1**) Line graph of CNN-LSTM. (**c2**) Scatter plot of CNN-LSTM. (**d1**) Line graph of ConvLSTM. (**d2**) Scatter plot of ConvLSTM. (**e1**) Line graph of STA-ResConvLSTM. (**e2**) Scatter plot of STA-ResConvLSTM.

**Table 1 sensors-23-08863-t001:** Statistics about the two main cities.

Parameters	Beijing			Tianjing		
	Min	Max	Mean	Min	Max	Mean
AQI $\left(\mu g\cdot m^{-3}\right)$	8	500	69.92	9	500	73.87
PM_2.5_	1	742	39.15	1	365	44.43
PM_2.5__24h	2	238	38.78	3	264	44.38
PM_10_	1	8733	72.52	1	3077	72.01
PM_10__24h	2	1790	71.33	3	977	71.71
SO_2_	1	43	3.06	1	334	8.85
SO_2__24h	1	14	3.07	2	31	8.85
NO_2_	1	144	29.20	1	157	35.22
NO_2__24h	2	101	29.19	3	128	35.26
O_3_	1	322	61.81	1	333	67.44
O_3__24h	2	322	103.73	7	333	117.53
CO $\left(mg\cdot m^{-3}\right)$	0.1	4	0.59	0.1	5.8	0.83
CO_24h	0.1	2.8	0.60	0.1	2.9	0.83
Temperature (K)	254.42	313.7	287.46	254.25	311.97	287.11
Dew point (K)	240.33	301.9	277.52	237.35	303.9	277.78
Sensible temperature (K)	247.80	318.09	286.96	247.45	316.92	286.25
Min temperature (K)	250.04	310.47	285.42	252.14	311.36	286.38
Max temperature (K)	254.79	314.8	288.79	254.45	312.49	287.86
Pressure (kPa)	986	1048	1012.60	993	1046	1016.53
Humidity (%)	6	100	54.73	3	100	57.66
Wind speed (miles/h)	0.02	8.59	1.91	0	19	3.18
Wind direction (°)	0	360	180.47	0	360	177.66
Cloudiness (%)	0	100	40.91	0	100	17.49
Weather id	500	804	778.65	500	804	755.66

**Table 2 sensors-23-08863-t002:** Correlation coefficients of air pollutants between Beijing and neighboring cities.

City Pair	AQI	PM_2.5_	PM_10_	SO_2_	CO	NO_2_	CO
Beijng and Baoding	0.674	0.671	0.560	0.589	0.589	0.628	0.589
Beijng and Cangzhou	0.509	0.505	0.570	0.443	0.450	0.512	0.440
Beijng and Chengde	0.663	0.680	0.566	0.430	0.618	0.684	0.618
Beijng and Datong	0.494	0.453	0.520	0.332	0.336	0.574	0.416
Beijng and Langfang	0.765	0.786	0.621	0.549	0.726	0.693	0.726
Beijng and Qinhuangdao	0.554	0.614	0.533	0.558	0.406	0.563	0.427
Beijng and Tangshan	0.631	0.656	0.563	0.398	0.468	0.611	0.468
Beijng and Tianjin	0.618	0.632	0.597	0.434	0.520	0.640	0.520
Beijng and Zhangjiakou	0.589	0.548	0.415	0.468	0.525	0.524	0.527

**Table 3 sensors-23-08863-t003:** Model parameters.

Layer Name		Output Size	Parameters	Values
STA-ResNet		24 × 10 × 32	(filter, channel, channel) × number of layers	(3 × 3, 8/16/32) × 1
	SAM		-	-
	CAM		-	-
			(filter, channel, channel) × number of layers	(3 × 3, 8/16/32) × 1
ConvLSTM		24 × 10 × 64	(filter, channel, channel) × number of layers	(3 × 3, 64) × 1
Full connected layer		256 × 1	layer nodes × number of layers	256 × 1
		10 × 1		10 × 1
-		-	Dropout	0.5
-		-	Batch size	128
-		-	Learning rate	0.0001
-		-	Epoch	50

**Table 4 sensors-23-08863-t004:** Performance evaluation indicators for model single-step prediction.

Models	RMSE	MAE	R^2^	IA
CNN	13.90	8.51	0.8166	96.03%
LSTM	12.23	7.57	0.8606	96.89%
CNN-LSTM	11.59	6.33	0.8897	97.42%
ConvLSTM	11.03	6.39	0.9036	97.74%
STA-Res ConvLSTM	9.82	5.86	0.9307	98.29%

Note: window size = 3; model performance evaluation indicators (RMSE, MAE, R^2^, and IA) are the predictors for the next 1 h.

**Table 5 sensors-23-08863-t005:** Comparison of multi-step prediction performance (window size = 8, forecast horizon = 1–6 h).

Models	RMSE	MAE	R^2^	IA
CNN	17.08	10.28	0.7207	94.08%
LSTM	16.55	9.76	0.7339	94.21%
CNN-LSTM	14.85	9.32	0.7726	95.16%
ConvLSTM	14.43	8.81	0.8214	95.88%
STA-Res ConvLSTM	12.63	8.52	0.8871	97.19%

**Table 6 sensors-23-08863-t006:** Testing error for model prediction.

Models	RMSE				MAE			
	1–12 h	13–24 h	25–36 h	37–48 h	1–12 h	13–24 h	25–36 h	37–48 h
CNN	18.97	21.13	22.16	23.29	11.98	13.51	13.99	14.52
LSTM	18.06	20.11	21.39	22.74	11.12	12.92	13.66	14.52
CNN-LSTM	15.49	16.65	17.22	19.45	9.56	10.55	11.70	13.28
ConvLSTM	15.23	16.81	17.10	19.00	9.51	10.81	11.35	12.37
STA-Res ConvLSTM	11.88	13.12	13.58	14.37	7.82	8.24	8.60	9.71

Note: window size = 48; prediction error is averaged out by model testing errors (RMSE and MAE) for the next t~t+n hours.

**Table 7 sensors-23-08863-t007:** Testing accuracy for model prediction.

Models	R^2^				IA			
	1–12 h	13–24 h	25–36 h	37–48 h	1–12 h	13–24 h	25–36 h	37–48 h
CNN	0.6317	0.5197	0.4295	0.3019	91.75	89.99	88.26	84.48
LSTM	0.6608	0.5369	0.4635	0.4167	92.91	90.71	89.28	88.03
CNN-LSTM	0.7826	0.7145	0.7091	0.6359	95.22	94.07	93.71	92.60
ConvLSTM	0.7949	0.7214	0.6976	0.6463	95.41	94.02	93.58	92.26
STA-Res ConvLSTM	0.8919	0.8658	0.8395	0.8161	97.37%	96.79	96.18	95.83

Note: window size = 48; model testing accuracies (R^2^ and IA) are the average of the prediction accuracy for the next t~t+n hours.

## Data Availability

Not applicable.

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
