# Peer review of "Prediction of Pollutant Concentration Based on Spatial–Temporal Attention, ResNet and ConvLSTM"

_sensors, 2023, doi:10.3390/s23218863_

Round 1
Reviewer 1 Report
Comments and Suggestions for Authors
This work presents a unique air quality prediction model that claims improvement over previous work. I write claims because the authors do not compare their work with previous studies. This work has potential for publication but must address major concerns before I accept it on my part. Below are details.
Major comment:
1. The manuscript MUST be sent to a professional English proofreading agency, and I expect a rewrite in many sentences during the next round.
2. Health-related literature review must be added to the introduction to establish the need for this work.
3. Air quality models estimate concentrations of air pollutants and can also predict air quality but with very short time spans. The authors have not distinguished between both possibilities, and this should be made clear in the manuscript.
4. It is not clear how far in the future the STA-ResConvLSTM model predicts! And the association between errors and longer periods.
5. There is no discussion section, and the discussion in the result section is limited and lacks comparison with previous work.
Specific Comments:
1. Line 36: Define PM2.5
2. Please go through the manuscript and subscript 2.5 in PM2.5 à PM2.5
3. Line 33: please add the health aspect of PM2.5. Why do we care about PM2.5?
4. Line 38: Not enough information about PM2.5 biases and why this is important
5. Line 39: Research methods for what? Please be precise! Hence: PM2.5 is estimated using air quality models or predicted using…
6. Line 41: Air quality models mostly estimate and somewhat predict! So which is it? You need to make this clear, and you need to distinguish between both cases.
7. Lines 45-56: Here, I am still confused! Are you trying to predict future PM2.5 or estimate PM2.5 values using the models? If you are trying to predict, please add how far in the future can these air quality models predict.
8. Lines 63-66 “Machine learning models ….. meteorology” Incomplete sentences, please rewrite.
9. Line 90: Define CNN
10. Line 96: change “Can enhancing the accuracy” to “Can enhance the accuracy”
11. Line 102: What is the “s” between Ding [35] and designed? Should be removed?
12. Line 145: remove the period after “effective.”
13. Line 189: change “the PM2.5 concentration will exceed the that's first intermediate limit of” To “the PM2.5 concentration will exceed the first intermediate limit of”
14. Line 191: change “sensitive; In 2021, the level of PM2.5 will exceed 75 ?? ? 3 13.01% of the time, which” To “sensitive. In 2021, the level of PM2.5 exceeded 75 ??/?3 on 13.01% of the time, which”
15. The text in figures 2, 3 and 4 is small.
16. Line 258: Please rewrite the following sentence “The ConvLSTM is the following layer., which implements spatiotemporal feature extraction.” I do not know what the authors are trying to say
17. Line 379: This journal requires a Discussion section. Do the authors mean “Results and Discussion”?
18. Line 389-401: Section 4.2 should be moved to the Methods sections.
19. Line 413: change “and ConvLSTM. this means that” To “and ConvLSTM. This means that”
20. Line 434: Does R2 also decrease? Or does it increase to show an improvement? I guess the authors should write down the change between values (before and after) for clarity.
Comments on the Quality of English LanguageThe manuscript MUST be sent to a professional English proofreading agency, and I expect a rewrite in many sentences during the next round.
Author Response
Response to Comments on the Manuscript:
“Title”
()
October 10, 2023
-------------------------------------------------------------------------------------------------------
The authors gratefully acknowledge the editors and the anonymous reviewers for their constructive comments. We have made a comprehensive revision for our previous manuscript. Specially, any revisions are highlighted using the "Track Changes" function in Microsoft Word. Please refer to the point by point response. Thank you for your time.
Response to comments by Reviewer #1:
We would like to gratefully thank the reviewer for his/her constructive comments and recommendations for improving the paper. A point-by-point response to the interesting comments raised by the reviewer follows.
Major comment:
Point 1. The manuscript MUST be sent to a professional English proofreading agency, and I expect a rewrite in many sentences during the next round.
Response 1: Thank you for your suggestion. We agree with the reviewer. We've done a round of touching up thesis sentences. Due to time constraints, we will send the paper to a professional organization for touch-ups if needed.
Point 2. Health-related literature review must be added to the introduction to establish the need for this work.
Response 2: Thank you for your suggestion. We agree with the reviewer. I have made changes in the text.
Point 3. Air quality models estimate concentrations of air pollutants and can also predict air quality but with very short time spans. The authors have not distinguished between both possibilities, and this should be made clear in the manuscript.
Response 3: Thank you for your suggestion. We agree with the reviewer. I have made changes in the text.
Point 4. It is not clear how far in the future the STA-ResConvLSTM model predicts! And the association between errors and longer periods.
Response 4: Thank you for your suggestion. We agree with the reviewer. I have made changes in the text.
Point 5. There is no discussion section, and the discussion in the result section is limited and lacks comparison with previous work.
Response 5: Thank you for your suggestion. We agree with the reviewer. I have made changes in the text.
Specific comments:
Point 1. Line 36: Define PM2.5.
Response 1: Thank you for your suggestion. I have made changes in the text.
Point 2. Please go through the manuscript and subscript 2.5 in PM2.5 à PM2.5.
Response 2: Thank you for your suggestion. I have made changes in the text.
Point 3. Line 33: please add the health aspect of PM2.5. Why do we care about PM2.5?
Response 3: Thank you for your suggestion. I have made changes in the text.
Point 4. Line 38: Not enough information about PM2.5 biases and why this is important.
Response 4: Thank you for your suggestion. I have made changes in the text.
Point 5. Line 39: Research methods for what? Please be precise! Hence: PM2.5 is estimated using air quality models or predicted using…
Response 5: Thank you for your suggestion. I have made changes in the text.
Point 6. Line 41: Air quality models mostly estimate and somewhat predict! So which is it? You need to make this clear, and you need to distinguish between both cases.
Response 6: Thank you for your suggestion. I have made changes in the text.
Point 7. Lines 45-56: Here, I am still confused! Are you trying to predict future PM2.5 or estimate PM2.5 values using the models? If you are trying to predict, please add how far in the future can these air quality models predict.
Response 7: Thank you for your suggestion. I have made changes in the text.
Point 8. Lines 63-66 “Machine learning models ….. meteorology” Incomplete sentences, please rewrite.
Response 8: Thank you for your suggestion. I have made changes in the text.
Point 9. Line 90: Define CNN.
Response 9: Thank you for your suggestion. I have made changes in the text.
Point 10. Line 96: change “Can enhancing the accuracy” to “Can enhance the accuracy”.
Response 10: Thank you for your suggestion. I have made changes in the text.
Point 11. Line 102: What is the “s” between Ding [35] and designed? Should be removed?
Response 11: Thank you for your suggestion. I have made changes in the text.
Point 12. Line 145: remove the period after “effective.”
Response 12: Thank you for your suggestion. I have made changes in the text.
Point 13. Line 189: change “the PM2.5 concentration will exceed the that's first intermediate limit of” To “the PM2.5 concentration will exceed the first intermediate limit of”
Response 13: Thank you for your suggestion. I have made changes.
Point 14. Line 191: change “sensitive; In 2021, the level of PM2.5 will exceed 75 ?? ? 3 13.01% of the time, which” To “sensitive. In 2021, the level of PM2.5 exceeded 75 ??/?3 on 13.01% of the time, which”
Response 14: Thank you for your suggestion. I have made changes.
Point 15. The text in figures 2, 3 and 4 is small.
Response 15: Thank you for your suggestion. I have made changes in the text.
Point 16. Line 258: Please rewrite the following sentence “The ConvLSTM is the following layer., which implements spatiotemporal feature extraction.” I do not know what the authors are trying to say
Response 16: Thank you for your suggestion. I have made changes in the text.
Point 17. Line 379: This journal requires a Discussion section. Do the authors mean “Results and Discussion”?
Response 17: Thank you for your suggestion. We have removed ‘°’.
Point 18. Line 389-401: Section 4.2 should be moved to the Methods sections.
Response 18: Thank you for your suggestion. I have made changes in the text.
Point 19. Line 413: change “and ConvLSTM. this means that” To “and ConvLSTM. This means that”
Response 19: Thank you for your suggestion. I have made changes.
Point 20. Line 434: Does R2 also decrease? Or does it increase to show an improvement? I guess the authors should write down the change between values (before and after) for clarity.
Response 20: Thank you for your suggestion. I have made changes in the text.
Please refer to the discussion part of the revised manuscript.
Due to time constraints, only the description of language errors in the article will be corrected. We will touch up the article in all aspects next time if necessary.

Reviewer 2 Report
Comments and Suggestions for Authors
General comments
The authors used the spatial-temporal big data correlation to predict the air pollutant concentration. Considering the correlation between different parameters, e.g., PM2.5, CO, NO2, etc., the correlation between them within a large area is easy to infer from the surrounding area. However, the correlation within a local area, i.e., local pollutant sources, impacts only a small region, without influencing the large region surrounding it. Therefore, the correlation between them is very small. In this case, the proposed method has limited applications. As the authors mentioned in the limitation of the model in the “Conclusion and Future” section, urban area (e.g., Beijing) usually has a large amount of pollution sources, compared with surrounding rural areas. Even in the urban area, things become much more complicated due to different sources of pollutants.
The authors mentioned that “ConvLSTM can extract the spatial–temporal correlation characteristics of the pollutant information more well” in the “Conclusions and Future” section. It’s better to explain it concisely.
Did the authors do data quality control before processing the data? I noticed in Figures 2-4 that there are some abnormal data points. If the data quality is bad, authors should eliminate them. If they are good, authors should give some explanation and how these abnormal data points influence the air pollutant forecast.
The authors did not mention how they used the data. How many data are used for training, validation, and prediction, respectively? And why?
Figure and Table captions: The captions for almost all the figures and tables are too simple, and not sufficient to explain the figures and tables. I recommend authors to add more information.
Figures 2, 3, and 4: the panels are too small to see them clearly. The figure captions do not mention the meaning of each panel, even if it’s shown in each panel. In addition, it’s not necessary to show all the data points. Instead, authors can show daily, and monthly averages.
Figure 10: The RMSE and MAE in panels a and b increase as the prediction time steps increase. As the authors mentioned “As shown in Figure 10, all prediction models' accuracy declines as the prediction time step grows.” Explain it.
Figure 11: revise the caption.
Specific comments
P2L50-53: consider citing more accurate references. For example, cite the following paper for WRF-Chem.
Grell, G. A., Peckham, S. E., Schmitz, R., McKeen, S. A., Frost, G., Skamarock, W. C., & Eder, B. (2005). Fully coupled “online” chemistry within the WRF model. Atmospheric environment, 39(37), 6957-6975. https://doi.org/10.1016/j.atmosenv.2005.04.027
P4L166-167: “The first step is to fill in the missing values in the dataset by simple linear interpolation.” How many data are missing in the dataset? Showing this information will indicate how much accuracy the linear interpolation is.
P4L171-173: “The equation below describes how the Min-Max standardization translates data values into the range [0,1]:” Is this step for normalization?
3. Methodology
P8L275-277: “As a result, the attention mechanism's main focus is to look for the inside significance of the original content, which it does by disregarding irrelevant data and giving more weight to significant data” How did authors do it?
4. Results
P12L380: Did the authors tune these model parameters? If not, why did the authors set parameters like this?
P13L402: The authors only showed the prediction performance. What about the training and validation steps?
P18L563: “extraction of” to “extract”
P18L569-570: “To better extract geographical information, one method is to grid pollution and meteorological data from several cities.” Hard to understand it. Rephrase it.
Comments on the Quality of English LanguageOk
Author Response
Response to Comments on the Manuscript:
“Title”
()
October 10, 2023
-------------------------------------------------------------------------------------------------------
The authors gratefully acknowledge the editors and the anonymous reviewers for their constructive comments. We have made a comprehensive revision for our previous manuscript. Specially, any revisions are highlighted using the "Track Changes" function in Microsoft Word. Please refer to the point by point response. Thank you for your time.
Response to comments by Reviewer #2:
We would like to gratefully thank the reviewer for his/her constructive comments and recommendations for improving the paper. A point-by-point response to the interesting comments raised by the reviewer follows.
General comment:
Point 1. The authors used the spatial-temporal big data correlation to predict the air pollutant concentration. Considering the correlation between different parameters, e.g., PM2.5, CO, NO2, etc., the correlation between them within a large area is easy to infer from the surrounding area. However, the correlation within a local area, i.e., local pollutant sources, impacts only a small region, without influencing the large region surrounding it. Therefore, the correlation between them is very small. In this case, the proposed method has limited applications. As the authors mentioned in the limitation of the model in the “Conclusion and Future” section, urban area (e.g., Beijing) usually has a large amount of pollution sources, compared with surrounding rural areas. Even in the urban area, things become much more complicated due to different sources of pollutants.
Response 1: Thank you for your suggestion. We agree with the reviewer.
Point 2. The authors mentioned that “ConvLSTM can extract the spatial–temporal correlation characteristics of the pollutant information more well” in the “Conclusions and Future” section. It’s better to explain it concisely.
Response 2: Thank you for your suggestion. We agree with the reviewer. I have added an explanation in section 5.
Point 3. Did the authors do data quality control before processing the data? I noticed in Figures 2-4 that there are some abnormal data points. If the data quality is bad, authors should eliminate them. If they are good, authors should give some explanation and how these abnormal data points influence the air pollutant forecast.
Response 3: Thank you for your suggestion. Anomalous data points are called mutation points in this paper. As can be seen in Figures 2-4, the total number of mutation points accounts for a small percentage of the total data. This mostly reflects the issue that there aren't many samples at the mutation locations in the overall dataset, which causes an issue with unequal data distribution. The occurrence causes the issue of inadequate learning in the prediction models, which makes it challenging to learn the pattern of change in pollutant concentration in the event of mutation. Because of this, certain models might be challenging to fit when there is an abrupt increase in pollutant concentration. This paper is explained in Section 4-4, the fourth point of the final summary.
Point 4. The authors did not mention how they used the data. How many data are used for training, validation, and prediction, respectively? And why?
Response 4: Thank you for your suggestion. We agree with the reviewer. I have added an explanation in section 2.2.
Point 5. Figure and Table captions: The captions for almost all the figures and tables are too simple, and not sufficient to explain the figures and tables. I recommend authors to add more information.
Response 5: Thank you for your suggestion. We agree with the reviewer. I have made changes.
Point 6. Figures 2, 3, and 4: the panels are too small to see them clearly. The figure captions do not mention the meaning of each panel, even if it’s shown in each panel. In addition, it’s not necessary to show all the data points. Instead, authors can show daily, and monthly averages.
Response 6: Thank you for your suggestion. We agree with the reviewer. I have made changes. This section is to demonstrate that pollutant and meteorological data are cyclical, which can also be described as time-dependent, so a year's worth of data is used in this paper to make a graphic presentation.
Point 7. Figure 10: The RMSE and MAE in panels a and b increase as the prediction time steps increase. As the authors mentioned “As shown in Figure 10, all prediction models' accuracy declines as the prediction time step grows.” Explain it.
Response 7: Thank you for your suggestion. We agree with the reviewer. I have added an explanation in section 4.3.
Point 8. Figure 11: revise the caption.
Response 8: Thank you for your suggestion. I have made changes.
Specific comments:
Point 1. P2L50-53: consider citing more accurate references. For example, cite the following paper for WRF-Chem.
Grell, G. A., Peckham, S. E., Schmitz, R., McKeen, S. A., Frost, G., Skamarock, W. C., & Eder, B. (2005). Fully coupled “online” chemistry within the WRF model. Atmospheric environment, 39(37), 6957-6975. https://doi.org/10.1016/j.atmosenv.2005.04.027
Response 1: Thank you for your suggestion. I have made changes.
Point 2. P4L166-167: “The first step is to fill in the missing values in the dataset by simple linear interpolation.” How many data are missing in the dataset? Showing this information will indicate how much accuracy the linear interpolation is.
Response 2: Thank you for your suggestion. I have made changes.
Point 3. P4L171-173: “The equation below describes how the Min-Max standardization translates data values into the range [0,1]:” Is this step for normalization?
Response 3: Thank you for your suggestion. I have made changes.
Point 4. P8L275-277: “As a result, the attention mechanism's main focus is to look for the inside significance of the original content, which it does by disregarding irrelevant data and giving more weight to significant data” How did authors do it?
Response 4: Thank you for your suggestion. I have made changes.
Point 5. P12L380: Did the authors tune these model parameters? If not, why did the authors set parameters like this?
Response 5: Thank you for your suggestion. I have made changes.
Point 6. P13L402: The authors only showed the prediction performance. What about the training and validation steps?
Response 6: Thank you for your suggestion. In deep learning prediction, the training and validation sets are better than the test set, all the results of the training and validation sets are not put here. The following table shows the results of single-step prediction for each model in the training set.
|
Models |
RMSE |
MAE |
R2 |
IA |
|
CNN |
13.37 |
8.07 |
0.8313 |
96.33% |
|
LSTM |
11.95 |
7.52 |
0.8751 |
97.20% |
|
CNN-LSTM |
9.33 |
5.99 |
0.9333 |
98.45% |
|
ConvLSTM |
9.17 |
5.91 |
0.9321 |
98.46% |
|
STA-Res ConvLSTM |
8.82 |
5.85 |
0.9407 |
98.58% |
Point 7. P18L563: “extraction of” to “extract”
Response 7: Thank you for your suggestion. I have made changes.
Point 8. P18L569-570: “To better extract geographical information, one method is to grid pollution and meteorological data from several cities.” Hard to understand it. Rephrase it.
Response 8: Thank you for your suggestion. I have made changes.
Please refer to the discussion part of the revised manuscript.
Due to time constraints, only the description of language errors in the article will be corrected. We will touch up the article in all aspects next time if necessary.

Round 2
Reviewer 1 Report
Comments and Suggestions for Authors
The discussion is too short and does not include a comparison with previous studies. Also, go back to the point-by-point response and paste your changes (your updated text) in each point. It is not useful for me that the author says “Thank you for your suggestion. I have made changes in the text.” I cannot find some of these changes in the manuscript or where the text has been addressed. I am still going to consider these major changes until I understand what the authors have changed.
Comments on the Quality of English LanguageI will check on the next round
Author Response
The authors gratefully acknowledge the editors and the anonymous reviewers for their constructive comments. We have made a comprehensive revision for our previous manuscript. Specially, any revisions are highlighted using the "Track Changes" function in Microsoft Word. Please refer to the point by point response. Thank you for your time.
Response to comments by Reviewer #1:
We would like to gratefully thank the reviewer for his/her constructive comments and recommendations for improving the paper. A point-by-point response to the interesting comments raised by the reviewer follows.
Major comment:
Point 1. The manuscript MUST be sent to a professional English proofreading agency, and I expect a rewrite in many sentences during the next round.
Response 1: Thank you for your suggestion. We agree with the reviewer. I've touched up the language in the article.
Please refer to the revised manuscript.
Point 2. Health-related literature review must be added to the introduction to establish the need for this work.
Response 2: Thank you for your suggestion. We agree with the reviewer. I have made changes in the text.
Please see references 4、5 and 6 in the manuscript.
Point 3. Air quality models estimate concentrations of air pollutants and can also predict air quality but with very short time spans. The authors have not distinguished between both possibilities, and this should be made clear in the manuscript.
Response 3: Thank you for your suggestion. We agree with the reviewer. I have made changes in the text.
Air quality modeling can estimate concentrations of air pollutants and can also predict air quality, but over short time spans. In this paper, air quality modeling is applied to the field of pollutant concentration prediction.
Please refer to the section introduction of the revised manuscript.
Point 4. It is not clear how far in the future the STA-ResConvLSTM model predicts! And the association between errors and longer periods.
Response 4: Thank you for your suggestion. We agree with the reviewer. I have made changes in the text.
The STA-ResConvLSTM model performs well in predicting pollutant concentrations at a time step of 48h with less fluctuation in accuracy, which suggests that the model can continue to predict pollutant concentrations for longer time periods. But as the prediction step increases, the accuracy of the prediction decreases.
Please refer to the section 4.4 of the revised manuscript.
Point 5. There is no discussion section, and the discussion in the result section is limited and lacks comparison with previous work.
Response 5: Thank you for your suggestion. In this paper, four deep learning models are used as comparison models. The comparison and justification of the past models have been fully discussed in sections 4.1, 4.2, and 4.3 of this paper, respectively, and the discussion section is just as a summary.
Please refer to the section 5 of the revised manuscript.
Specific comments:
Point 1. Line 36: Define PM2.5.
Response 1: Thank you for your suggestion. I have defined PM2.5.
Please refer to the section introduction of the revised manuscript.
Point 2. Please go through the manuscript and subscript 2.5 in PM2.5 à PM2.5.
Response 2: Thank you for your suggestion. We have changed all PM2.5 to PM2.5.
Please refer to the revised manuscript.
Point 3. Line 33: please add the health aspect of PM2.5. Why do we care about PM2.5?
Response 3: Thank you for your suggestion. I have made changes in the text.
Please see references 4 and 5 in the manuscript.
Point 4. Line 38: Not enough information about PM2.5 biases and why this is important.
Response 4: Thank you for your suggestion. Here is an error in the presentation of this article. The general idea here is that accurate prediction of PM2.5 concentrations is important. This has been carefully revised.
Please refer to the first paragraph of the section introduction of the manuscript
Point 5. Line 39: Research methods for what? Please be precise! Hence: PM2.5 is estimated using air quality models or predicted using…
Response 5: Thank you for your suggestion. Here is an error in the presentation of this article. The general idea here is that PM2.5 concentration prediction can be viewed as a time-series processing problem that can be predicted based on past historical correlation data, e.g., meteorological factors such as temperature and humidity, as well as other pollution factors such as PM10 and O3.
Please refer to the second paragraph of the section introduction of the manuscript
Point 6. Line 41: Air quality models mostly estimate and somewhat predict! So which is it? You need to make this clear, and you need to distinguish between both cases.
Response 6: Thank you for your suggestion. "Evaluation" is a misrepresentation. What the text is trying to convey is that pollutant concentration predictions. Here is an error in the presentation of this article. The general idea here is that numerical model-based prediction methods and data-driven model-based prediction methods. The first is the prediction method of numerical modeling that simulates the process of emission, diffusion, transformation and removal of air pollutants through meteorological principles and statistical methods so as to achieve the prediction of pollutant concentrations
Please refer to the second paragraph of the section introduction of the manuscript
Point 7. Lines 45-56: Here, I am still confused! Are you trying to future PM2.5 or estimate PM2.5 values using the models? If you are trying to predict, please add how far in the future can these air quality models predict.
Response 7: Thank you for your suggestion. I have made changes in the text. "Evaluation" is a misrepresentation. What the text is trying to convey is that pollutant concentration predictions. The concept of short-term forecasting of pollutants has been added to the text
Please refer to the second paragraph of the section introduction of the manuscript
Point 8. Lines 63-66 “Machine learning models ….. meteorology” Incomplete sentences, please rewrite.
Response 8: Thank you for your suggestion. I have made changes in the text.
Machine learning models, an important type of artificial intelligence learning, combine the trends of the pollutants themselves and the intrinsic relationship between the pollutants and meteorology to produce predicted concentration values.
Please refer to lines 79 through 82 of the manuscript
Point 9. Line 90: Define CNN.
Response 9: Thank you for your suggestion. I have made changes in the text.
Please refer to lines 107 of the manuscript
Point 10. Line 96: change “Can enhancing the accuracy” to “Can enhance the accuracy”.
Response 10: Thank you for your suggestion. I have made changes in the text.
Please refer to lines 112 of the manuscript
Point 11. Line 102: What is the “s” between Ding [35] and designed? Should be removed?
Response 11: Thank you for your suggestion. I have made changes in the text.
Please refer to lines 112 of the manuscript
Point 12. Line 145: remove the period after “effective.”
Response 12: Thank you for your suggestion. I have made changes in the text.
Please refer to lines 118 of the manuscript
Point 13. Line 189: change “the PM2.5 concentration will exceed the that's first intermediate limit of” To “the PM2.5 concentration will exceed the first intermediate limit of”
Response 13: Thank you for your suggestion. I have made changes.
Please refer to lines 209 of the manuscript
Point 14. Line 191: change “sensitive; In 2021, the level of PM2.5 will exceed 75 ?? ? 3 13.01% of the time, which” To “sensitive. In 2021, the level of PM2.5 exceeded 75 ??/?3 on 13.01% of the time, which”
Response 14: Thank you for your suggestion. I have made changes.
Please refer to lines 211-213 of the manuscript
Point 15. The text in figures 2, 3 and 4 is small.
Response 15: Thank you for your suggestion. I have made changes in the text. The word in the chart has been changed to #36
Please refer to figures 2, 3 and 4 of the manuscript
Point 16. Line 258: Please rewrite the following sentence “The ConvLSTM is the following layer., which implements spatiotemporal feature extraction.” I do not know what the authors are trying to say
Response 16: Thank you for your suggestion. I have made changes in the text.
Please refer to lines 278-279 of the manuscript
Point 17. Line 379: This journal requires a Discussion section. Do the authors mean “Results and Discussion”?
Response 17: Thank you for your suggestion. This paper is discussed extensively in the last part of 4.4.
Point 18. Line 389-401: Section 4.2 should be moved to the Methods sections.
Response 18: Thank you for your suggestion. I have made changes in the text.
Point 19. Line 413: change “and ConvLSTM. this means that” To “and ConvLSTM. This means that”
Response 19: Thank you for your suggestion. I have made changes.
Please refer to lines 433 of the manuscript
Point 20. Line 434: Does R2 also decrease? Or does it increase to show an improvement? I guess the authors should write down the change between values (before and after) for clarity.
Response 20: Thank you for your suggestion. I have made changes in the text.
Please refer to the section 4.4 of the revised manuscript.

Round 3
Reviewer 1 Report
Comments and Suggestions for Authors
NA
Comments on the Quality of English LanguageNA